# Magic cancellation point for vibration resilient ultrastable microwave signal synthesis

**William Loh** ✉, **Dodd Gray, Ryan Maxson, Dave Kharas, Jason Plant, Paul W. Juodawlkis, Cheryl Sorace-Agaskar** & **Siva Yegnanarayanan**

Photonically-synthesized microwave signals have surpassed the phase-noise performance achievable by traditional means of RF signal generation. However, for microwave-photonic oscillators to truly replace their RF counterparts, this phase-noise advantage must also be realizable when operating outside of a laboratory. Oscillators are known to be notoriously vibration sensitive, with both traditional RF and optical oscillators degrading sharply in all but the most stationary of environments. We demonstrate here a powerful technique that makes use of a precise frequency difference between two optical signals, termed the magic cancellation point, to suppress the vibration-induced noise upon optical frequency division to the RF. We showcase the cancellation of vibration noise by 22.6 dB, achieving an acceleration sensitivity of $1.5 \times 10^{-10} \, \text{g}^{-1}$. Beyond mitigating the effects of vibration, this technique also preserves the excellent phase noise obtained by optical frequency division and reaches $-72$ dBc/Hz and $-139$ dBc/Hz at 10 Hz and 10 kHz offset frequencies on a 10 GHz carrier. This technique applies widely to optical carriers of any center wavelength and derived from an arbitrary resonator geometry.

Oscillators are pivotal in a vast array of applications that span from consumer electronics to basic science. Two prevalent types of oscillators, optical and RF oscillators, have traditionally coexisted in separate application spaces, that share little overlap due to the three or more orders of magnitude separation in their operating frequencies. However, the recent development of techniques for optical frequency division (OFD) has opened up an exciting possibility for optical sources to replace RF sources by coherently dividing down an optical carrier to microwave frequencies[1–10]. Owing to their higher quality factors[11–15], optical sources generally outperform their RF counterparts by several orders of magnitude, and could revolutionize the capabilities of systems that are conventionally constructed purely from RF electronics, such as radar systems, radio astronomy telescopes, advanced communications networks, high-end analog-to-digital converters, and precision test equipment.

Despite the capacity of optical oscillators for lower noise, a key question remains whether this performance advantage would translate into tangible benefits in a real-world setting. In particular, the resilience of an oscillator to vibration plays an important role in determining an oscillator's overall phase noise – and low size, weight, and power (SWaP) quartz crystal oscillators form the backbone of nearly all electronics equipment in existence today. Comparing an optical reference with an instability of $1 \times 10^{-13}$ to quartz with an instability of $1 \times 10^{-10}$ and given an acceleration sensitivity of $-1 \times 10^{-9} \, \text{g}^{-1}$ for both systems, the performance of the two would be indistinguishable under an acceleration of 0.1g. Moreover, the relative performance of the two would be preserved in the division to microwave frequencies, and most of the performance gains of starting with an optical carrier would be lost under modest levels of vibration.

Recently, techniques of optical frequency division using a partial-spanning comb have garnered significant interest due to their inherent simplicity and amenability to chip integration while also retaining the potential for excellent phase noise[6–10]. One large benefit of such techniques is that the baseline for frequency division is based on the

MIT Lincoln Laboratory, Lexington, MA, USA. ✉e-mail: william.loh@ll.mit.edu

frequency difference between two optical carriers, which enables a rare opportunity for substantial common-mode noise rejection (CMNR) when both carriers are derived from a common cavity[10,16]. However, it should be understood that the CMNR increases at the same rate that the division factor decreases due to their opposite scaling with the comb span, and thus the product of the division ratio and CMNR always remains constant. This constrains the acceleration sensitivity of the synthesized microwave after division to still be limited to that of the optical source, despite acceleration being largely common-mode to the optical cavity.

Here, we showcase a method to overcome this tradeoff between CMNR and division ratio which importantly enables the divided-down microwave signal to outperform the original optical carrier in acceleration sensitivity by an order of magnitude or more. We refer to this method as the magic cancellation point, named after an analogous effect in atomic physics, termed the magic wavelength, in which the AC Stark shift between two states cancels to zero at a specific laser wavelength[17,18]. In our scheme, this cancellation point for vibration occurs at a precise value of frequency separation between the two optical carriers used as the baseline for division. Starting with two stimulated Brillouin scattering (SBS) lasers[19-23] separated by a baseline of ~1 THz, we perform division down to 10 GHz and achieve 22.6 dB cancellation of the vibration-induced phase noise to reach an acceleration sensitivity of $1.5 \times 10^{-10}\,g^{-1}$. This vibration suppression comes without sacrificing phase noise performance, and we demonstrate phase noise levels of −72 dBc/Hz and −139 dBc/Hz at 10 Hz and 10 kHz offset frequency, respectively. With further refinement to the depth of the vibration null, we anticipate this magic cancellation point to be a powerful tool for achieving vibration resilience in future microwave-photonic systems.

## Results

Our demonstration of a vibration canceled RF oscillator employs the core principles of performing frequency division using a partial-spanning frequency comb. The baseline for division is formed by anchoring the two endpoints of the frequency comb, a technique that we term optical difference frequency division (ODFD)[10]. This technique is more widely known as two-point optical frequency division[24], though we prefer our alternative naming scheme to make clear the potential for considerable CMNR when frequency differences are taken. Due to the wide bandwidths available in the optical domain, the subtraction of two optical signals yields a large baseline in the range of 1 THz or more. Upon division to 10 GHz, the phase noise improves by 40 dB, reaching levels competitive with the state-of-the-art in microwave oscillators. More importantly, however, if both of the comb's anchor points respond identically to external perturbations, then these perturbations effectively become invisible to the baseline and also to the resulting synthesized microwave signal.

Unfortunately, there is no guarantee for the two optical carriers that form the ODFD baseline to respond identically to vibration. One powerful strategy is to stabilize both signals to the same high quality factor (Q) resonator, accomplishing the tying of both signals to a common medium while simultaneously reducing their underlying phase noise. Yet, as the modes that the lasers are stabilized to will inevitably have different mode numbers, their response to vibration will also be slightly different. Furthermore, should two modes be chosen in close frequency proximity to one another, the result will be a deterioration of the baseline extent and also of the division factor. This represents the fundamental tradeoff between CMNR and division factor in ODFD, and as a consequence, the divided-down microwave signal can only equal, but not improve over, the acceleration sensitivity of the base optical carriers. We overcome this limitation by rotating the polarization of one of the lasers to the orthogonal axis. Though no exact cancellation point exists for distinct modes of the same mode family, the same is not generally true for modes belonging to different

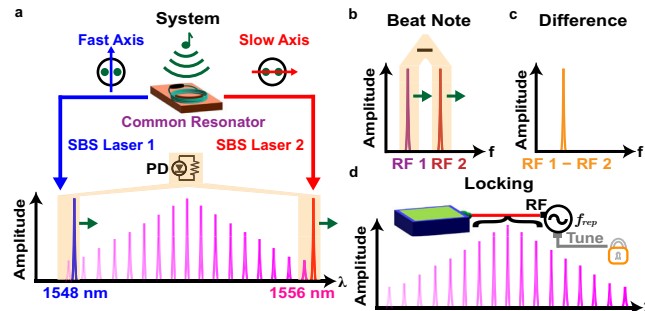

**Fig. 1 | Vibration cancellation concept. a** The vibration canceled ODFD system comprises SBS laser light generated along the two orthogonal polarization axes of fiber (slow and fast) and heterodyned against two comb lines. A vibration applied to the system causes the two SBS lasers to fluctuate in the same direction with equal magnitude. **b** The SBS lasers and the comb lines interfere to produce two separate beat notes at microwave frequencies. The beat notes fluctuate via the applied vibration, with the polarity of the movement dependent on which comb lines form the beat (shown here to be in the same direction). **c** The two RF beat notes are sent to a mixer, and the difference frequency is selected. The applied vibration is canceled in the resulting difference, thereby yielding no fluctuation of the RF tone. **d** The comb lines, formed by electro-optic modulation at a frequency $f_{rep}$, are stabilized to the RF difference frequency, suppressing the effects of vibration on the derived microwave signal.

families (see Methods Section ODFD Vibration Suppression). However, this cancellation only occurs for a specific baseline separation, and once the wavelength of one of the lasers is chosen, the other laser wavelength will also be decided.

Figure 1a shows our ODFD scheme that makes use of two stimulated Brillouin scattering (SBS) lasers generated from a common fiber resonator. The shorter (longer) wavelength laser is aligned to the fast (slow) axis, and both lasers are sent to heterodyne against the nearby lines of a frequency comb on a photodetector (PD). An applied perturbation to the system causes both lasers to shift in frequency, which results in corresponding shifts to the two beat notes generated at RF frequencies (Fig. 1b). For the configuration of comb lines shown in Fig. 1a, this shift in the RF beat notes occurs in the same direction. By mixing the two RF signals together and selecting the difference frequency (Fig. 1c), the applied perturbation precisely cancels in the resulting microwave tone. Finally, after locking the comb's repetition rate to keep this difference constant, the information of the SBS lasers' divided-down phase noise becomes imprinted on the repetition rate (Fig. 1d). The derived microwave signal ($f_{rep}$) and its associated noise ($\Delta f_{rep}$) are given by

$$f_{rep} + \Delta f_{rep} = \frac{(f_1 - f_2) + (\Delta f_1 - \Delta f_2)}{m} \quad (1)$$

where $f_1$ and $f_2$ represent the frequencies of lasers 1 and 2, $\Delta f_1$ and $\Delta f_2$ represent the frequency fluctuations of lasers 1 and 2 due to external perturbations, and $m$ is the division factor given by the total number of comb lines between the two lasers. Notably from Eq. (1), if the external perturbations are coherent to both lasers and equal in magnitude, then their impact on the divided-down microwave signal becomes negated.

The configuration of our ODFD system is shown in Fig. 2a. The system consists of three separate subsystems aimed at 1) generating the two orthogonal polarization ultranarrow-linewidth lasers, 2) creating an electro-optically modulated (EOM) frequency comb to span the spectrum of the two lasers[25], and 3) using electronics signal processing to perform optical frequency division to microwave frequencies. The two SBS lasers are formed by sending two amplified pump lasers into a 10-meter fiber SBS resonator created using a 95%/5% fiber-optic coupler. The pump lasers centered at 1556 nm and 1548 nm

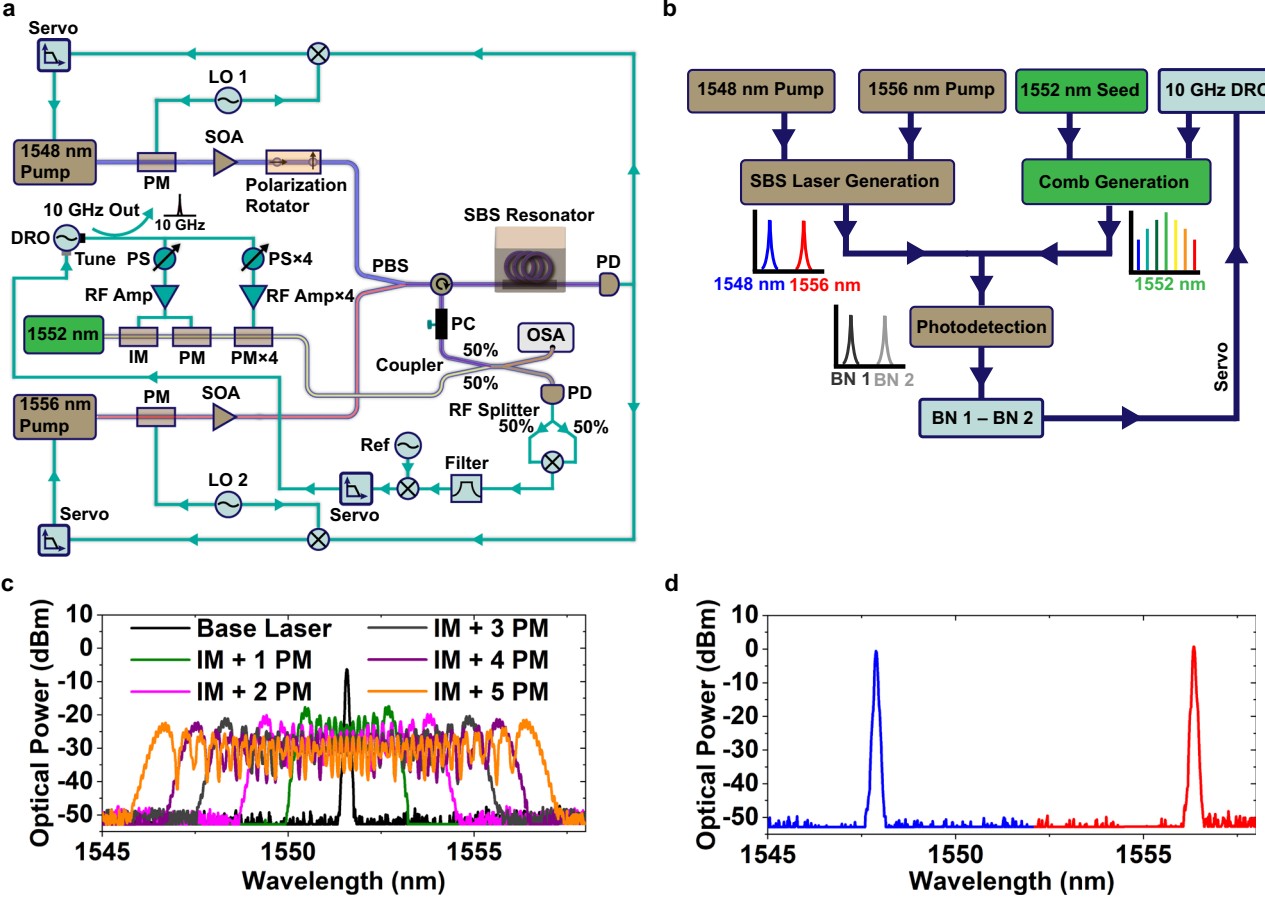

**Fig. 2 | Optical difference frequency division system. a** ODFD system diagram comprising two orthogonally polarized SBS lasers combined with an electro-optically modulated frequency comb. DRO dielectric resonator oscillator, PS phase shifter, LO local oscillator, IM intensity modulator, PM phase modulator, SOA semiconductor optical amplifier, PBS polarization beam splitter, PC polarization controller, OSA optical spectrum analyzer. The DRO is stabilized to the optical beat notes to perform division down to 10 GHz. **b** Flow diagram depicting the signal path from SBS pump lasers, EOM seed laser, and 10 GHz DRO driver to the stabilized frequency divider output. **c** EOM comb spectrum comprising successive stages of modulation. The comb span broadens with each stage of modulation up to a width of 11.5 nm for 1 intensity modulator and 5 phase modulator stages. **d** SBS laser spectrum with wavelength scale aligned to the plotted frequency comb spectrum. The two SBS lasers are orthogonally polarized and directly overlap with the comb lines.

are Pound-Drever-Hall (PDH)[26] locked to the resonator by phase modulating the pump light and subsequently photodetecting the light transmitted past the resonator. We use a wideband tunable laser as the 1548 nm pump to enable scanning for the exact vibration cancellation point. In addition, the 1548 nm pump light is rotated in polarization to the fast axis prior to being sent into the SBS resonator. The two orthogonal polarizations are combined via a polarization beam splitter, and the SBS output is extracted through a circulator.

The EOM comb subsystem is formed using a second tunable laser that undergoes a series of modulation stages. The modulators are driven by a dielectric resonator oscillator operating at 10 GHz, whose output is split multiple ways, each passing through a phase shifter and an RF amplifier. The first modulation stage comprises both an intensity and phase modulator, while subsequent stages (up to 4) provide additional phase modulation. The comb's center frequency is tunable via the frequency of its base laser, which we place at 1552 nm to span the gap of the two SBS lasers. The SBS lasers are combined with the comb output on a 3 dB coupler, and a polarization controller is used to ensure a fraction of both SBS lasers have a nonzero beat with the comb.

After photodetection on a high speed photodetector, the two beat notes (BN 1 and BN 2) are split equally two ways and then mixed together. The mixing products consist of the following: BN 1 − BN 1, BN 1 + BN 1, BN 2 − BN 2, BN 2 + BN 2, BN 1 − BN 2, and BN1 + BN 2. Only BN 1 − BN 2 and BN 1 + BN 2 are relevant for use in ODFD phase locking.

Depending on the polarity of the beat note between the nearest comb line and the SBS lasers, either the sum or difference frequency should be selected after mixing. The resulting RF signal has the common-mode vibration canceled and is then mixed with a reference oscillator to create a phase-locked loop that acts on the DRO frequency to keep the sum/difference constant. A portion of the stabilized DRO is coupled out as the 10 GHz ODFD output. The SBS laser subsystem is on a vibration platform capable of applying a controlled vibration along the three coordinate axes (x-horizontal, y-vertical, and z-out of plane). Particular attention is paid to negating the influence of vibration on all other components of the system beyond the SBS resonator (see Methods ODFD System Configuration). This ensures the ODFD system's vibration response to be nearly entirely determined by the SBS resonator and the degree of common-mode cancellation we achieve. A flow diagram depicting the signal pathway for accomplishing ODFD is shown in Fig. 2b

Figure 2c shows the EOM comb spectrum with successive stages of modulation. Each modulation stage broadens the comb by ~2 nm until a maximum width of 11.5 nm. The two SBS lasers have outputs that overlap strongly with the comb spectrum as is observed in Fig. 2d. We intentionally leave additional comb extent on the blue wavelength side to allow for scanning of the 1548 nm SBS laser to find the optimal vibration cancellation point. The SBS laser output powers are 0.9 mW and 1.2 mW at 1548 nm and 1556 nm, respectively, while the EOM comb

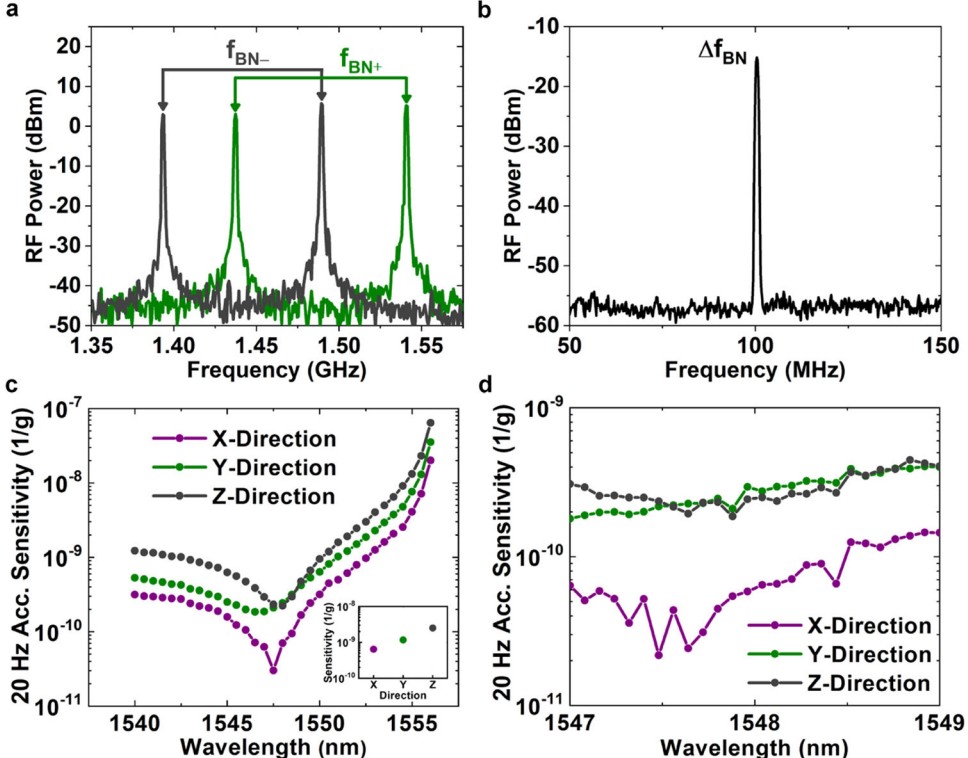

**Fig. 3 | Magic cancellation point measurements. a** Two instances of beat notes formed between the SBS lasers and their nearby comb lines. Noise is intentionally introduced into the comb's center frequency resulting in the beat notes moving in tandem to higher and lower frequencies. **b** After mixing the beat notes and choosing the difference frequency, the resulting microwave signal narrows to a single peak with the common-mode noise canceled. **c** Broadband scan of the magic cancellation point for an applied 20 Hz vibration to the SBS laser subsystem. For all three coordinate axes, the cancellation point converges to a narrow band of wavelengths around 1547.5 nm. The inset shows the 20 Hz acceleration sensitivity of the base SBS fiber laser. **d** Wavelength scan of the magic cancellation point near the acceleration minimum. The acceleration sensitivity improves over the base laser in all three axes.

outputs ~5 µW per line. Our use of a longer-length (10 m) fiber SBS resonator (see Methods Fiber SBS Laser Operation) enables a greatly increased resonator Q and a lower oscillator phase noise, but clamps the maximum SBS output power to ~1 mW before the next-order Stokes wave turns on. Furthermore, 10 meters also represents an upper bound on the practical operating length for an SBS resonator as the resulting mode spacing of 20 MHz becomes comparable to the Brillouin gain bandwidth in fiber ( ~50 MHz).

Two sets of beat notes are generated between the SBS lasers and their neighboring EOM comb lines. These beat notes can exhibit a high degree of common-mode fluctuations with noise present on either the SBS lasers or the comb lines. Figure 3a shows two example states of the beat notes when an intentional modulation is applied to the comb lines towards higher and lower frequencies. The beat notes individually fluctuate by ~500 MHz, yet their difference remains fixed at 100 MHz (Fig. 3b). The spectral width of the difference frequency also narrows from that of the beat notes themselves due to common-mode noise cancellation. Although, Fig. 3a and b only show the cancellation of noise solely on the comb lines, the more interesting scenario for the cancellation of vibration noise on the SBS lasers would be indistinguishable from the standpoint of the beat notes formed.

Perfect vibration cancellation, however, occurs only at a specific baseline span between the two SBS lasers, and Fig. 3c shows the resulting acceleration sensitivity when the shorter wavelength SBS laser is scanned while the longer wavelength SBS laser is kept constant. For these measurements, a 20 Hz sinusoidal tone is applied to the vibration platform that the SBS laser subsystem rests on. The applied vibration induces phase noise sidebands on the microwave beat notes, that become suppressed in their difference frequency. The measured residual sideband power is converted to a fractional frequency shift,

and then normalized to an independent measure of the applied vibration taken by an accelerometer (see Methods Vibration Calibration). Both the vibration table and the accelerometer allow for vibration to be applied and measured along the three (X, Y, and Z) coordinate axes, which enables us to track the global minimum of the magic cancellation point. From Fig. 3c, the minimum in acceleration sensitivity occurs near 1547.5 nm in all three axes and is highest in the Z-direction. The acceleration sensitivity quickly rises away from this minimum point, especially towards the longer wavelength side where a shorter baseline span results in a further degraded vibration performance.

Due to the precise nature of the magic cancellation point, it is necessary to sample the minimum at 1547.5 nm with a precision finer than 0.5 nm. Figure 3d shows a sweep from 1547 nm to 1549 nm of the measured acceleration sensitivity sampled at every comb line (0.08 nm spacing). As an added procedure, we complete the system lock at each point and directly measure the acceleration sensitivity of the final 10 GHz microwave output. As the acceleration sensitivity of the derived 10 GHz microwave (Fig. 3d) agrees with that of the optical difference frequency (Fig. 3c), the optical frequency division process is observed to precisely transfer the vibration cancellation to the stable microwave output. The minimum in acceleration sensitivity occurs for the Z-direction at 1547.9 nm, reaching an acceleration sensitivity of $1.9 \times 10^{-10}\,\mathrm{g}^{-1}$. Compared to the base SBS laser shown in the inset of Fig. 3c with a sensitivity of $2.5 \times 10^{-9}\,\mathrm{g}^{-1}$, the stable 10 GHz microwave improves on the vibration performance by a factor of 13.5 (22.6 dB in phase noise). For the X and Y directions, the 10 GHz microwave reaches acceleration minima of $2.2 \times 10^{-11}\,\mathrm{g}^{-1}$ and $1.8 \times 10^{-10}\,\mathrm{g}^{-1}$, respectively. Across the spectrum, the acceleration sensitivity is consistently largest in the Z direction. We believe this to be due to the resonator sitting on

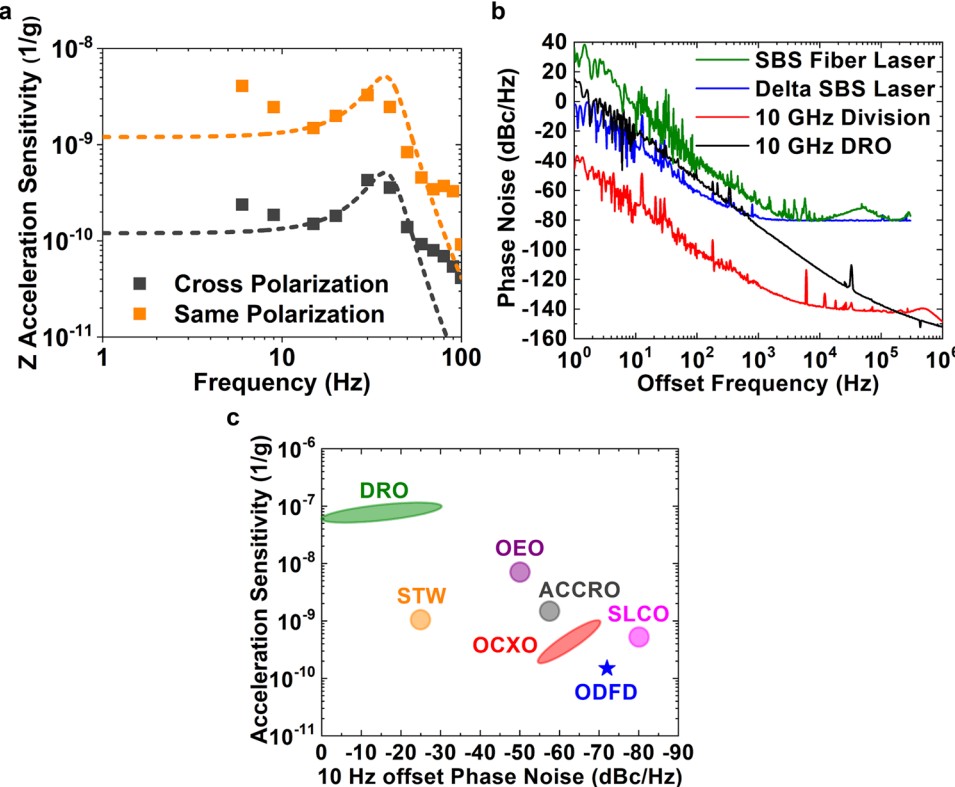

**Fig. 4 | Optical difference frequency division measurements. a** Measured frequency response of the ODFD system against applied vibration in the Z direction. The response is compared for an ODFD system having two SBS lasers operating with the same and cross polarization states.. A second order response is fit to each of the measured vibration response functions. **b** Measured phase noise of the 10 GHz ODFD system compared to the phase noise of the base SBS lasers, the difference frequency between the SBS lasers, and the 10 GHz DRO oscillator. **c** Comparison of phase noise and acceleration sensitivity across several free-running microwave oscillator technologies at 10 GHz. STW surface transverse wave, OEO optoelectronic oscillator; ACCRO air-dielectric ceramic-cavity resonator oscillator, OCXO oven controlled crystal oscillator, SLCO sapphire loaded cavity oscillator. Our phase-locked ODFD performance is also plotted.

the vibration platform along the Z axis. Table vibrations directly translate to perturbations of the resonator, whereas the other two directions do not have a direct point of contact to the table. Furthermore, the asymmetry between X and Y is likely a result of the positioning and anchoring of the optical fibers that exit the resonator enclosure.

It is interesting to measure the acceleration sensitivity of the ODFD system across a spectrum of frequencies. We perform this measurement in the Z-axis where the acceleration sensitivity was found to be largest and with the ODFD system operated at the magic cancellation point. The input sinusoidal drive is varied from 6 Hz to 100 Hz, and the acceleration sensitivity of the 10 GHz ODFD output is measured and calibrated against an accelerometer at each point (Fig. 4a). Both lower and higher frequencies were tested beyond the plotted range, but as the response of the vibration platform is not uniform across the spectrum, these frequencies yielded phase noise sidebands below the measurable phase noise floor of the 10 GHz ODFD output. Nevertheless, we observe an acceleration response that resembles an underdamped second-order system with a natural frequency of 40 Hz, as is indicated by the fits to the measured spectra. At the magic cancellation point and using orthogonal polarization lasers, the Z-axis acceleration sensitivity reaches $1.5 \times 10^{-10}\,g^{-1}$ at 15 Hz. As a test of our vibration cancellation technique, we modify our ODFD system to use two SBS lasers of the same polarization state (slow-axis aligned) to form the baseline. Under these conditions, the CMNR is still significant ( ~45 dB) but cannot yield perfect cancellation. As a result, the CMNR and division factor together only transfers the vibration sensitivity of the base SBS laser onto the 10 GHz microwave. This is observed in Fig. 4a where the measured Z acceleration sensitivity is

$2.0 \times 10^{-9}\,g^{-1}$ at 20 Hz, comparable to that of the base SBS laser shown in the inset of Fig. 3c. Though the shape of Fig. 4a comprises additional features beyond the response of a simple second order system, our vibration cancellation technique uniformly suppresses the ODFD acceleration sensitivity by an order of magnitude across the spectrum.

Figure 4b shows the phase noise of the optical frequency divider at various points along the system. The noise of the base SBS fiber laser is measured through its heterodyne beat with an Er-doped fiber frequency comb locked to an ultralow expansion cavity. An integration over the SBS laser phase noise yields an integrated linewidth[27] of 18.6 Hz. The noise features beyond 10 kHz offset frequency are not inherent to the SBS fiber laser itself, and are instead due to the reference laser through the combination of the ULE cavity servo and the Er frequency comb lock. After taking the difference between a 1548 nm SBS laser and a 1556 nm SBS laser via the EOM comb, common-mode noise becomes suppressed. Beyond the cancellation of effects of vibration, fundamental thermorefractive noise that limits the fiber SBS laser also becomes greatly reduced resulting in lower phase noise. Due to their levels, we note that the remaining spurs in the difference frequency spectrum can be inferred to be noise that is not correlated between the two SBS lasers, and thus would not be reduced by common-mode noise suppression. One possible uncorrelated noise source could be the added electrical noise present on each of the independent PDH servos.

Our ODFD system acts on the SBS laser difference and divides the result to a microwave frequency of 10 GHz. Starting from a baseline of 1.05 THz between the two SBS lasers, the division factor is 105 × for a total phase noise reduction of 40.4 dB. From Fig. 4b, we observe that the 10 GHz divided output is ~40 dB lower than the SBS laser

difference, as is expected from ideal division. At 1 Hz, 10 Hz, and 100 Hz offset frequencies, the 10 GHz phase noise reaches phase noise levels of −42 dBc/Hz, −72 dBc/Hz, and −102 dBc/Hz, respectively. For these low offset frequencies, the phase noise follows a slope of −30 dB/decade, which we believe to be still limited by thermorefractive noise. Though common-mode thermal noise is greatly attenuated in the SBS difference, the level of suppression is finite and presents a phase noise floor to the 10 GHz divided output. This low-frequency noise may be further reduced by using larger resonator mode volumes or greater baseline division factors. Beyond 1 kHz offset frequency, the SBS laser difference becomes limited by additive white noise, while the 10 GHz divided output continues to decrease further in phase noise to a floor of −142 dBc/Hz. The measured phase noise of the base 10 GHz DRO used for EOM drive is also shown for reference. The phase noise is consistently ~50 dB above the 10 GHz ODFD output except at higher offset frequencies, where the two converge past 1 MHz. The servo bump results in a slight increase in noise of the 10 GHz divided output near the ~500 kHz DRO phase locked loop bandwidth.

Figure 4c showcases the phase noise and acceleration sensitivity of the ODFD system and of a variety of microwave oscillators in existence today. Note that the microwave oscillators all operate free-running, which cannot be strictly compared against the phase-locked operation of the ODFD. Nevertheless, with these differences in mind, it is still useful to put the ODFD system performance in the context of commonly used microwave oscillators. For the purposes of uniformity, phase noise is compared scaled to a center frequency of 10 GHz and at an offset frequency of 10 Hz, while the acceleration sensitivity is compared across oscillators using the axis most sensitive to acceleration. In addition, we do not consider oscillators that use active means of vibration sensing and compensation. In general, acceleration data is sparse for microwave oscillators outside of the ubiquitously-used quartz crystal oscillators, and we instead rely on a few key reports of measurements provided in literature[28,29]. We find our ODFD operating at the magic cancellation point to outperform other oscillator varieties in terms of acceleration sensitivity while simultaneously achieving excellent phase noise performance across the spectrum.

## Discussion

We anticipate the magic cancellation point to be a powerful tool in providing operational robustness to microwave signals derived from frequency division of an optical carrier. Despite the already exceptional performance demonstrated here, more and larger improvements should be possible pending photonic integration of the ODFD system. Our SBS fiber resonator used here has many degrees of vibrational freedom, which may have contributed to the overall smearing of the depth of the acceleration null at the magic cancellation point. We expect system integration on chip to greatly limit the degrees of freedom for motion, potentially increasing the suppression of vibration beyond the 22.6 dB achieved here. At the same time, we also expect the system's acceleration sensitivity to further improve on chip, providing a lower base level of response to acceleration to which the cancellation is applied. Finally, chip integration brings with it the possibility to engineer the waveguide cross sections, which may enable placement of the magic cancellation point to arbitrary wavelengths and baseline spans. The ability to choose the point of cancellation may be especially important for future ODFD systems as currently the magic cancellation point for vibration does not overlap with the cancellation point for temperature fluctuations in our fiber SBS resonator. However, should it be possible to overlap both cancellation points, a vibration and temperature insensitive microwave synthesizer may be achievable. Such a scheme is also expected to improve the ODFD's current low-frequency thermorefractive noise limitations.

Our demonstration of the existence of a magic cancellation point where vibration becomes suppressed showcases the potential that photonics offers to microwave systems. Having access to a bandwidth

that is 20,000 × larger, a baseline formed by differences in optical signals enables 22.6 dB cancellation of acceleration-induced phase noise while still being able to retain exceptional noise performance on a 10 GHz carrier. Furthermore, the acceleration suppression of the magic cancellation point uniformly reduces the acceleration response across the spectrum of the base laser with no sign of saturation, suggesting that further advancements are possible with improvements to the vibration resiliency of the base laser and/or to the degree of vibration suppression via the magic cancellation point. We expect future photonic integration of the ODFD system to greatly alter the landscape of what is possible, not only in terms of acceleration suppression, but also in system size, power, manufacturability, cost, and portability.

## Methods
### Optical difference frequency division vibration suppression
In this section, we derive the central equations that govern frequency division of two lasers into the microwave frequency range. The frequencies of the lasers are given by $f_1$ and $f_2$ and are each defined relative to the center frequency of a frequency comb ($f_{center}$) and an integer multiple of the comb line spacing ($f_{rep}$).

$$f_1 = f_{center} + kf_{rep} \tag{2}$$

$$f_2 = f_{center} - lf_{rep} \tag{3}$$

Here, $k$ and $l$ refer to the two integer number of comb lines that come in between the comb center and lasers $f_1$ and $f_2$, respectively. Taking $m = k + l$, it follows from Eqs. (2) and (3) that the difference frequency between the two lasers is given by

$$f_1 - f_2 = mf_{rep} \tag{4}$$

Upon solving for $f_{rep}$ and explicitly separating out the noise contributions of the generated microwave ($\Delta f_{rep}$), and of the two lasers, denoted as $\Delta f_1$ and $\Delta f_2$, we obtain

$$f_{rep} + \Delta f_{rep} = \frac{(f_1 - f_2) + (\Delta f_1 - \Delta f_2)}{m} \tag{5}$$

For uncorrelated noise contributions between the individual lasers, the power spectral densities of $\Delta f_1$ and $\Delta f_2$ add, and the net result becomes divided by $m^2$ upon conversion to a microwave carrier. This sets the noise floor of the ODFD output to be a fraction of the noise of the base lasers.

However, for correlated noise, which is the more interesting case regarding acceleration sensitivity, we start by relating $\Delta f_1$ and $\Delta f_2$ to a perturbation of the free spectral range (FSR) of the resonator that the lasers are generated from.

$$\Delta f_1 = i\Delta f_{FSR} \tag{6}$$

$$\Delta f_2 = j\Delta f_{FSR} \tag{7}$$

Here, $\Delta f_{FSR}$ is the perturbation in the resonator's free spectral range, and $i$ and $j$ are integers corresponding to the mode numbers of $f_1$ and $f_2$, respectively. Subtracting one from the other, we find that

$$\Delta f_1 - \Delta f_2 = (i - j)\Delta f_{FSR} = m\frac{f_{rep}}{f_{FSR}}\Delta f_{FSR} \tag{8}$$

The second relation is found after multiplying by $\frac{f_{FSR}}{f_{rep}} \times \frac{f_{rep}}{f_{FSR}}$ and recognizing that $m = (i-j)\frac{f_{FSR}}{f_{rep}}$. Finally, as $\frac{\Delta f_{FSR}}{f_{FSR}} = \frac{\Delta n}{n}$ where $n$ is the

refractive index, we find that

$$\Delta f_1 - \Delta f_2 = m f_{rep} \frac{\Delta n}{n} \qquad (9)$$

Using this result in Eq. (5) yields

$$\frac{\Delta f_{rep}}{f_{rep}} = \frac{\Delta n}{n} \qquad (10)$$

Thus, the fractional change in the microwave frequency is directly related to the fractional perturbation of the refractive index. As $\Delta n / n = \Delta f_1 / f_1 = \Delta f_2 / f_2$, we observe that the acceleration sensitivity of the microwave output remains unaltered from that of the base laser.

Implicit in these calculations is the assumption that the two lasers are generated from modes of the same mode family. The situation changes greatly if this restriction is lifted. In particular, we modify Eqs. (6) and (7) to account for the fact that the FSRs would be different for the two lasers.

$$\Delta f_1 = i \Delta f_{FSR,1} \qquad (11)$$

$$\Delta f_2 = j \Delta f_{FSR,2} \qquad (12)$$

As the modes are still part of the same resonator, $\Delta f_{FSR,1}$ and $\Delta f_{FSR,2}$ are still correlated with one another. The difference between $\Delta f_1$ and $\Delta f_2$ now yields

$$\Delta f_1 - \Delta f_2 = i \Delta f_{FSR,1} - j \Delta f_{FSR,2} \qquad (13)$$

In the situation above, it is possible to find two distinct modes whose subtraction yields identically zero, which we have termed the magic cancellation point. To find the condition for exact cancellation, we write out $\Delta f_{FSR,1}$ and $\Delta f_{FSR,2}$ in terms of refractive index perturbations due to a vibration event

$$\Delta f_{FSR,1} = -\frac{f_{FSR,1}}{n_1} \frac{dn_1}{da} \Delta a \qquad (14)$$

$$\Delta f_{FSR,2} = -\frac{f_{FSR,2}}{n_2} \frac{dn_2}{da} \Delta a \qquad (15)$$

where $dn_{1,2}/da$ is the change in refractive index of lasers 1 and 2 with respect to acceleration and $\Delta a$ is a perturbation in acceleration. Substituting Eqs. (14) and (15) into Eq. (13) yields

$$\Delta f_1 - \Delta f_2 = -i \frac{f_{FSR,1}}{n_1} \frac{dn_1}{da} \Delta a + j \frac{f_{FSR,2}}{n_2} \frac{dn_2}{da} \Delta a \qquad (16)$$

Setting $\Delta f_1 - \Delta f_2$ equal to 0 to solve for the magic cancellation point and recognizing that $i f_{FSR,1} = f_1$ and $j f_{FSR,2} = f_2$, we obtain

$$\frac{f_1}{n_1} \frac{dn_1}{da} = \frac{f_2}{n_2} \frac{dn_2}{da} \qquad (17)$$

Equation (17) represents the relationship between the acceleration sensitivities of the refractive indices required to operate at the magic cancellation point. A more intuitive way to interpret Eq. (17) can be achieved by solving for the frequency of laser 2 in terms of laser 1 at the point of cancellation

$$f_2 = f_1 \frac{n_2}{n_1} \frac{dn_1/da}{dn_2/da} \qquad (18)$$

To obtain the largest baseline for frequency division, the response of the individual lasers to acceleration must be as asymmetric as possible. In our work, we have experimentally determined the baseline to be 1.05 THz near 1550 nm when one laser operates on the slow axis mode of optical fiber and the other laser operates on the fast axis mode. From Eq. (18), this baseline equates to -0.54% asymmetry in the responses of $n_1$ and $n_2$ to acceleration.

Equation (18) also predicts that exact cancellation is possible and that the vibration phase noise theoretically can reach zero. However, in practice, we find that the magic cancellation point in optical fiber is broadened, likely by the many degrees of freedom in which fiber can vibrate and induce an index change (i.e. $dn/da$ is not a constant and depends slightly on acceleration strength and direction). We experimentally showcased vibration suppression of 22.6 dB, which indicates that the cancellation condition of Eq. (18) is satisfied to within 92.6%. Note that Eq. (18) predicts that multiple magic cancellation points exist for a given system, simply through variation of the operating wavelengths of the two lasers. However, for a fixed wavelength of laser 1, it is unlikely for more than one magic cancellation point to exist except in cases where the refractive index behavior varies non-monotonically.

## Optical difference frequency division system configuration

Many aspects of the ODFD system require significant care to enable cancellation of the system's acceleration. In principle, every element in the ODFD system is vibration sensitive. However, certain elements such as the pump lasers and the DRO are locked and greatly diminish the influence of vibration on their operation. This applies to the entire pathway of the pump light up to the point that the pump light converts to an SBS wave in the resonator. Note that for strong vibrations, the DRO may lose phase locking, and the DRO's acceleration sensitivity would then be an important consideration. A single photodetector is used to demodulate the PDH error signals for both pump lasers, which is intended make the laser paths as common-mode as possible. Other elements, such as function generators, operate at low frequencies relative to the -200 THz optical waves, and are far less susceptible to vibration. Note that all the function generators used here were operated independently from one another. However, these function generators may also be synchronized should they prove to be a limitation to the common-mode rejection. The comb system itself is also completely common to the beat notes with both SBS lasers, and thus center frequency shifts of the comb laser or of the comb pathway are exactly canceled when the difference frequency is taken. We have tested this by applying large-scale frequency modulation to the central laser of the EOM comb and saw no degradation in the ODFD phase noise.

The primary risk to canceling acceleration occurs once the SBS lasers are generated. After this point, any imbalance to the pathway fluctuations of each laser translates to noise that imperfectly cancels in their difference frequency. To prevent these effects, we intentionally engineer both SBS lasers to traverse a common pathway for light, up until the point that both lasers are photodetected. In particular, from the output of the resonator, both SBS lasers pass through a circulator and are routed to a 50/50 splitter that combines the comb light with the SBS laser light. The light is detected using a single photodetector, which generates two microwave tones corresponding to the beat notes with each of the two SBS lasers. At this point, the electrical pathway is split in two using two minimal and equal length cables, and the two paths are mixed together to yield the difference frequency. After this point, the difference frequency carries the information of the SBS lasers, and has a much lower sensitivity to vibration due to its low frequency. Thus, with the aforementioned design considerations applied to our system, the acceleration sensitivity of our ODFD system becomes dominated by that of the SBS resonator. As a result, we expect that our vibration measurements of the magic cancellation technique performed on the SBS laser subsystem applies generally to the whole system.

**Stimulated brillouin scattering laser and electro-optically modulated comb operation**

Our SBS resonator is formed from a 95/5 fiber coupler with two of its ends spliced together to form a 10-meter fiber coil. The two pump lasers are locked to the resonances of the cavity, and the corresponding generated SBS laser outputs are used as the anchor points for frequency division. The SBS lasers are ~10.8 GHz shifted to lower frequencies compared to their respective pump lasers. The mode spacing is 20 MHz, which is smaller than the SBS gain bandwidth of ~50 MHz. This gain bandwidth only specifies the point where the gain is attenuated by 3 dB, and we have verified that our current 20 MHz mode spacing yields a sufficient gain differential to enable stable single-mode operation of the SBS laser.

The EOM comb is intentionally operated with the seed laser frequency at the center between the two SBS laser outputs. To achieve the maximum comb span, the phases of the RF drive to each of the modulators need to be aligned for constructive interference. The exact phase required is dependent on the phase delay along both the RF and optical paths, and will be different system to system. As a consequence, these phases are manually tuned via analog control knobs while monitoring the comb extent on an optical spectrum analyzer.

**Vibration calibration**

We measure the acceleration sensitivity of our system by applying a sinusoidal modulation at a known frequency to a vibration table that the SBS laser subsystem rests on. First, the amount of frequency movement is found by measuring the corresponding phase noise sideband ($L(f)$) that is generated[30]. This sideband is next converted to a frequency fluctuation using the fact that phase and frequency are related by a derivative.

$$2f^2 L(f) = S_\nu(f) \tag{19}$$

Separately, an accelerometer is mounted to the vibration table and is used to measure the applied acceleration at the vibration frequency. Converting the power spectral density of frequency fluctuations to linear units, normalizing to the center frequency, and then dividing by the acceleration response yields the system's acceleration sensitivity. Note that this calibration procedure requires the measurements of phase noise and of the acceleration response to be performed with the same resolution bandwidth. Alternatively, integrated powers may instead be used with no restrictions on resolution bandwidth. To find the spectral response of the system's acceleration sensitivity, the applied vibration is swept across the desired frequency range.

## Data availability
The data sets that support this study are available upon request.

## Code availability
The code used for analysis and simulations are available upon request.

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

## Acknowledgements

DISTRIBUTION STATEMENT A. Approved for public release. Distribution is unlimited. This material is based upon work supported by the Under Secretary of Defense for Research and Engineering under Air Force Contract No. FA8702-15-D-0001. Any opinions, findings, conclusions or recommendations expressed in this material are those of the author(s) and do not necessarily reflect the views of the Under Secretary of Defense for Research and Engineering. The authors thank Octave Photonics for their assistance with setting up the EOM comb.

## Author contributions

W.L., S.Y. conceived, designed and carried out the experiments with the magic cancellation point. W.L., D.G. conceived, designed and carried out the experiments with the EOM comb. W.L. conceived, designed and carried out the experiments with the SBS laser. R.M. performed the resonator packaging. W.L., D.G., R.M., D.K., J.P., P. W.J., C.S. and S.Y. discussed the results and contributed to the manuscript.

## Competing interests

The authors declare no competing interests.
