## [Transparent Peer Review file · Nature Communications]

Magic Cancellation Point for Vibration Resilient Ultrastable Microwave Signal Synthesis

Corresponding Author: Dr William Loh

Version 0:

Reviewer comments:

Reviewer #1

(Remarks to the Author)

In this manuscript, a concept called “magic cancellation point” is proposed, which is used for vibration resilient ultrastable microwave signal synthesis. It is an interesting method. I have a few concerns, looking for response from the author.

- (1) The explanation of the magic cancellation point in the manuscript can demonstrate that frequency drift has been effectively suppressed. However, the effect of the magic cancellation point on phase noise is not intuitive enough. Please provide a detailed mathematical description of the relationship between the magic cancellation point and phase noise. In addition, please explain how this point will affect the phase noise limit.
- (2) What parameters in the system setup are related to the magic cancellation point? Is there a quantitative relationship? Will the magic cancellation point drift? Are there several magic cancellation points for the same system?
- (3) In the Fig. 4(b), the phase noise in the frequency offset range from 1Hz to 100kHz is improved by employing the proposed method. However, there are some spurs, such as spiking spurs in the frequency offset range from 10Hz to ~40Hz, near the frequency offset at 100Hz and 5kHz. Please explain their origin. It also can be seen that the phase noise in the frequency offset range from 2MHz to 10MHz is worse than that of the 10-GHz DRO, why?
- (4) It is better to provide a signal flow diagram to explain the scheme in Fig. 2.

Reviewer #2

(Remarks to the Author)

In the past few decades, microwave generation and frequency synthesis using photonics-based technologies, such as optical frequency division (OFD), optoelectronic oscillator (OEO), etc., have shown tremendous potential in achieving unprecedented phase noise performance that can even surpass the most advanced electrical counterparts. In particular, OFD-based microwave synthesis has recently attracted significant attention in both scientific and industrial fields due to its remarkable ability to coherently divide optical reference oscillators' frequency down to the RF domain, while preserving the ultra-high fractional frequency stability of the original optical reference oscillation. Such high performance microwave source can be used in a wide range of applications. However, many reported OFD systems so far have only been demonstrated in laboratory environments, which are free from vibrations. The viability, robustness, and susceptibility to phase noise degradation of OFD-based microwave oscillators operating under strong vibrational conditions, such as outside laboratory environments, remain inadequately investigated.

In this manuscript, William Loh and his colleagues explored a novel technique, the so-called “magic cancellation point”, to mitigate the phase noise acceleration sensitivity in a two-point OFD system where the two optical carriers are referenced to the same optical cavity. The authors break the conventional two-point OFD system paradigm that references both optical carriers to the distinct modes of the same mode family. They proposed an innovative scheme to reference the two SBS lasers to the different mode families of the same SBS resonator, that opens the possibility to break the constraint of the tradeoff between the common-mode noise rejection and the frequency division factor. Even though this happens at specific wavelengths and spacings between these two SBS lasers, such schemes can be generalized to OFD systems using different optical reference cavities. I can see the great potential of such an idea in photonic integrated OFD systems to reduce the influence of vibration on the RF phase noise. The authors showcase up to 22.6 dB phase noise cancellation without phase noise degradation under strong vibration conditions for a generated 10 GHz microwave signal, achieved when the two SBS lasers are spaced by ~1 THz. The overall manuscript is well organized and well written. The details of the system are well elaborated. Moreover, this topic is of broad interest, and the overarching concept presented in this

manuscript is expected to inspire future research on photonic integrated OFD systems with enhanced acceleration sensitivity. Therefore, I recommend the publication of this manuscript in Nature Communications after some minor revisions.

1. In the ODFD system shown in Fig. 2, is the SBS resonator temperature-stabilized? Additionally, does the "magic cancellation point" remain effective under strong temperature fluctuations of the SBS resonator? If this is the case, I believe it would significantly simplify the OFD system by eliminating the need for temperature stabilization.
2. In the implementation of SBS lasers, fiber cavities of 10 meters in length are used, resulting in a mode spacing of 20 MHz, while the typical SBS gain bandwidth is 50 MHz. I am wondering whether such a long fiber might induce mode hopping in the SBS lasers under vibration conditions. If this occurs, it could alter the two beat notes produced by the beating of the SBS lasers and their nearest comb lines, which might lead to the failure of the phase locking of DRO.
3. From Fig. 4b, the OFD system exhibits relatively high phase noise at offset frequencies below 10 Hz, which I believe is primarily dominated by the SBS fiber lasers. Could the authors provide further details on the noise sources of the SBS lasers in this frequency range and discuss potential methods for noise reduction?
4. Given the application of the magic cancellation point, to what extent can enclosing the SBS resonator within a vibration-isolation housing further enhance phase noise cancellation under strong vibration conditions?
5. What is the locking bandwidth of the PLL used to lock the DRO in the OFD system?
6. From the measurement results in Fig. 3, why is the acceleration sensitivity highest in the Z-direction and lowest in the X-direction? What factors might contribute to this behavior?
7. Since the authors used a DRO to synthesize the 10 GHz signal, and Fig. 4c shows that the DRO exhibits relatively poor acceleration sensitivity, I wonder whether the DRO might lose phase locking under strong vibration conditions. If this is the case, an RF oscillator with reduced vibration sensitivity would also be necessary in the OFD system to enhance robustness under strong vibration conditions.

Reviewer #3

(Remarks to the Author)

This work proposes a method for achieving maximum suppression of vibration noise in microwave signal synthesis using optical frequency division (ODFD) and the principle of "magic cancellation point". The authors demonstrated a 22.6 dB vibration noise cancellation effect through theoretical derivation and systematic experimental verification, maintaining excellent phase noise performance at a carrier frequency of 10 GHz. Overall, this manuscript is innovative and persuasive in terms of experimental verification, and the theme has a potentially significant impact on microwave photonics, precision measurement, and communication systems. However, several significant issues related to the fundamental principle, experimental design, and conclusion need to be addressed prior to acceptance.

Issue 1:

The manuscript emphasizes that only the SBS laser subsystem is placed on the vibration platform, while the entire ODFD system is not tested under the same vibration environment. Therefore, the evaluation of the system's overall vibration resilience is limited. How would the full system behave under vibration? Would it be possible to achieve the same order of magnitude in vibration suppression for the complete setup? Is there a realistic potential for full-system vibration-resilient applications in the future?

Issue 2:

In the PDH locking process, the authors should clarify whether the phase modulation signals (LO1 and LO2) are derived from the same local oscillator source. Based on my knowledge, operating two PDH loops with independent LO sources may degrade the ultimate common-mode noise rejection and could introduce additional differential-mode noise.

Issue 3:

In the EOM subsystem, the DRO output signal is distributed to multiple phase modulators via several phase shifters. The authors should provide detailed descriptions of the phase shifts applied to each branch because the proper phase alignment is critical for ensuring the uniformity of the comb teeth generated by the EOM system.

Issue 4:

The identification of the "magic cancellation point" in the manuscript is primarily based on experimental observation. However, the theoretical analysis related to common-mode noise cancellation is overly simplified. The manuscript lacks supporting theoretical modeling and simulation to validate the underlying mechanism.

Issue 5:

From the experimental results shown in Figure 3, it is evident that the vibration cancellation along the Z-axis is the least effective. The authors should explain why the Z-axis exhibits the worst cancellation performance compared to the X- and Y-axis.

Issue 6:

In Figure 4, the phase noise data is presented only under static conditions without applied vibration. The manuscript lacks a comparative measurement of phase noise under vibration, both with and without magic cancellation applied. Such a comparison is necessary to demonstrate the effectiveness of the magic cancellation mechanism in improving the final signal quality under realistic vibration conditions.

Issue 6:

According to Figure 2, two beat notes from a high-speed photodetector are split equally into two ways and then mixed

together. The structure is confusing because the two signals injected into the RF mixer are identical and the sum or difference frequency cannot be obtained at output of the mixer. Clarifications or corrections should be made to this part.

Issue 7:

The author should add key vibration parameters in the acceleration sensitivity measurement, such as the vibration power spectral density or the acceleration value.

Issue 8:

In this paper, the proposed system is phase locked to an RF reference using the ODFD method. The comparison of phase noise and acceleration sensitivity in Fig. 4(c) is not appropriate, as the oscillators provided in literature [28] operate in a free-running state without any phase-locking loop. This discrepancy could result in incorrect conclusions, and the comparison should be carefully revised.

Version 1:

Reviewer comments:

Reviewer #1

(Remarks to the Author)

This manuscript proposes an innovative microwave signal synthesis method. The author has effectively addressed all the issues that concern me. I suggest that this manuscript can be accepted.

Reviewer #2

(Remarks to the Author)

The authors have thoroughly addressed all of my concerns. I recommend the manuscript in its current form for publication in Nature Communications.

Reviewer #3

(Remarks to the Author)

The issue that was raised has now been effectively resolved, and I believe that it aligns with the publishing requirements.

We want to express our gratitude to the reviewers for dedicating their time to a complete and thorough review of the manuscript. We feel the reviewers did an excellent job assessing the manuscript's merits and deficiencies and sincerely believe that their comments have greatly strengthened the overall quality of the reported work. We address each respective reviewer's comments below.

Reviewer 1:

1. *In this manuscript, a concept called “magic cancellation point” is proposed, which is used for vibration resilient ultrastable microwave signal synthesis. It is an interesting method. I have a few concerns, looking for response from the author.*

We thank the reviewer for their kind words and their careful review of our manuscript.

2. *The explanation of the magic cancellation point in the manuscript can demonstrate that frequency drift has been effectively suppressed. However, the effect of the magic cancellation point on phase noise is not intuitive enough. Please provide a detailed mathematical description of the relationship between the magic cancellation point and phase noise. In addition, please explain how this point will affect the phase noise limit.*

We thank the reviewer for identifying this deficiency in the current manuscript. We have added these details to the Methods Section A. Equations (14)-(18) have been added to derive the condition for exact cancellation of the vibration noise, in which the vibration-induced phase noise can theoretically be completely cancelled. In practice, we experimentally find 22.6 dB suppression and have provide a paragraph to explain the discrepancy from the ideal theory. For space saving reasons, we do not duplicate the added derivations and analysis here.

3. *What parameters in the system setup are related to the magic cancellation point? Is there a quantitative relationship? Will the magic cancellation point drift? Are there several magic cancellation points for the same system?*

We think this question is partly related to the previous reviewer question and may be a result of our incomplete theory on the magic cancellation point at the time of manuscript submission. We believe that Eqs. (14)-(18) that we added addresses this point. The reviewer also raises an additional interesting question related to whether there are there are several magic cancellation points for the same system. Because the exact cancellation point is dependent on the wavelength of the second laser relative to the chosen wavelength for first laser (see. Eq. (18)), in principle many cancellation points can exist by simply varying the wavelength of the two lasers while satisfying Eq. (18). Perhaps another interesting question to ask is if multiple cancellation points exist assuming the wavelength of the first laser is fixed. Such a scenario is also possible, but is unlikely except in cases when the behavior of the refractive index across wavelength varies non-monotonically. We have added the following discussing to Methods A ODFD Vibration Suppression.

“To obtain the largest baseline for frequency division, the response of the individual lasers to acceleration must be as asymmetric as possible. In our work, we have experimentally determined the baseline to be 1.05 THz near 1550 nm when one laser operates on the slow axis mode of optical fiber and the other laser operates on the fast axis mode. From Eq. (18), this baseline equates to $\sim 0.54\%$ asymmetry in the responses of n_1 and n_2 to acceleration.

Equation (18) also predicts that exact cancellation is possible and that the vibration phase noise theoretically can reach zero. However, in practice, we find that the magic cancellation point in optical fiber is broadened, likely by the many degrees of freedom in which fiber can vibrate and induce an index change (i.e. dn/da is not a constant and depends slightly on acceleration strength and direction). We experimentally showcased vibration suppression of 22.6 dB, which indicates that the cancellation condition of Eq. (18) is satisfied to within 92.6%. Note that Eq. (18) predicts that multiple magic cancellation points exist for a given system, simply through variation of the operating wavelengths of the two lasers. However, for a fixed wavelength of laser 1, it is unlikely for more than one magic cancellation point to exist except in cases where the refractive index behavior varies non-monotonically.”

4. *In the Fig. 4(b), the phase noise in the frequency offset range from 1Hz to 100kHz is improved by employing the proposed method. However, there are some spurs, such as spiking spurs in the frequency offset range from 10Hz to ~ 40 Hz, near the frequency offset at 100Hz and 5kHz. Please explain their origin. It also can be seen that the phase noise in the frequency offset range from 2MHz to 10MHz is worse than that of the 10-GHz DRO, why?*

This is an astute observation by the reviewer regarding the phase-noise performance of our optical difference frequency divider. Regarding the spurs in the 10 Hz to 40 Hz range, this is not a particularly easy question to answer. We have a few clues however that can narrow the range of possibilities. 1) The divided down microwave spectrum matches the shape of the spectrum corresponding to the difference frequency between the two SBS lasers. This means that the division process is simply bringing the noise that already exists on the difference frequency between the two SBS lasers down by the ~ 40 dB division factor rather than adding new noise onto the spectrum. 2) The spur frequencies do not match commonly seen line noise harmonics found in the range of 60 Hz, 120 Hz, 180 Hz, etc. 3) The spurs do not match anything seen on the DRO spectrum, and thus cannot be attributed to the microwave source. 4) Comparing the delta SBS laser spectrum to the measured noise of a single SBS laser, it can be inferred that the spurs do not occur due to a noise source common to the optical cavity (e.g., vibration, temperature, etc). This conclusion can be reached by observing that the ratio of the peaks to background noise does not remain constant between the individual spectra. Furthermore, calculating the amount of suppression expected for common-mode noise from the mode number difference and the additional magic cancellation point suppression: $20\log(1.05 \text{ THz}/193.4 \text{ THz}) - 22.6 \text{ dB} = -67.9 \text{ dB}$, we find that this suppression can not account for the spur heights observed. Finally, upon zooming in further on the plotted phase noise (see Fig. R1 of the reviewer response letter), we find that the peaks of the SBS difference frequency spectra do not correspond in frequency to any spurs observed in the spectrum of a single SBS laser. This suggests that these spurs are simply hidden underneath the phase noise of the SBS lasers.

With these clues, we can make an educated guess that these peaks are likely due to electrical noise on each of the independent PDH locks. We think that while the above explanation may be very interesting to the reviewer, it may be a bit too detailed and can be a confusing or distracting point for an average reader. We instead summarize the conclusions for the reader in the main text of manuscript:

“Due to their levels, we note that the remaining spurs in the difference frequency spectrum can be inferred to be noise that is not correlated between the two SBS lasers, and thus would not be reduced by common-mode noise suppression. One possible uncorrelated noise source could be the added electrical noise present on each of the independent PDH servos.”

Figure R1: Zoom-in on Figure 4b of the main manuscript

Regarding the reviewer’s second question of why the phase noise between 200 kHz-2 MHz is worse than the 10 GHz DRO phase noise – that fortunately has a more straightforward explanation. To perform the optical frequency division, a servo is used to lock the DRO multiplied up by the electro-optic frequency comb to the optical difference between the two SBS lasers. This servo, while reducing the noise at frequencies lower than the servo bandwidth, will add noise in the direct vicinity of the servo bandwidth (termed the servo bump). It arises due to the phase accumulation at high offset frequencies resulting in a 180 degree phase near the servo bandwidth. This phase shift results in positive feedback rather than negative feedback and accumulates noise at these frequencies. However, as the gain continues to rolloff past the servo bandwidth, the noise reverts to the noise of the underlying DRO oscillator. We have added the following to the paragraph describing Fig. 4b:

“The servo bump results in a slight increase in noise of the 10 GHz divided output near the DRO phase locked loop bandwidth (~ 500 kHz).”

5. *It is better to provide a signal flow diagram to explain the scheme in Fig. 2.*

We thank the reviewer for pointing this out. We have provided a signal flow diagram in Fig. 2b and rearranged the rest of the figure ordering.

Reviewer 2:

1. *In this manuscript, William Loh and his colleagues explored a novel technique, the so-called “magic cancellation point”, to mitigate the phase noise acceleration sensitivity in a two-point OFD system where the two optical carriers are referenced to the same optical cavity. The authors break the conventional two-point OFD system paradigm that references both optical carriers to the distinct modes of the same mode family. They proposed an innovative scheme to reference the two SBS lasers to the different mode families of the same SBS resonator, that opens the possibility to break the constraint of the tradeoff between the common-mode noise rejection and the frequency division factor. Even though this happens at specific wavelengths and spacings between these two SBS lasers, such schemes can be generalized to OFD systems using different optical reference cavities. I can see the great potential of such an idea in photonic integrated OFD systems to reduce the influence of vibration on the RF phase noise. The authors showcase up to 22.6 dB phase noise cancellation without phase noise degradation under strong vibration conditions for a generated 10 GHz microwave signal, achieved when the two SBS lasers are spaced by ~ 1 THz. The overall manuscript is well organized and well written. The details of the system are well elaborated. Moreover, this topic is of broad interest, and the overarching concept presented in this manuscript is expected to inspire future research on photonic integrated OFD systems with enhanced acceleration sensitivity. Therefore, I recommend the publication of this manuscript in Nature Communications after some minor revisions.*

We thank the reviewer for their kind words and greatly appreciate their careful review of our manuscript.

2. *In the ODFD system shown in Fig. 2, is the SBS resonator temperature-stabilized? Additionally, does the “magic cancellation point” remain effective under strong temperature fluctuations of the SBS resonator? If this is the case, I believe it would significantly simplify the OFD system by eliminating the need for temperature stabilization.*

The SBS resonator is temperature stabilized via a thermistor and TEC. The point about the magic cancellation point applying to temperature is very perceptive. Indeed, this was one of the first things we had checked too, since the effects of vibration and temperature on refractive index are very similar. From our tests, it appears that the magic cancellation point for vibration, at least in our system, does not currently apply to temperature. Temperature itself appears to have a separate magic cancellation point, but the two points do not overlap. We agree with the reviewer that if these two points were to overlap, or engineered to be at the same spot, that would be an enormous benefit to the OFD architecture. We think the many degrees of freedom

that fiber can move in response to vibration may have contributed to the two cancellation points being offset from one another. We have updated the Supplementary Information Section with the following:

“The enclosure is temperature controlled via a thermistor and thermoelectric cooler to a setpoint near room temperature.”

We have also updated the manuscript with the following:

“The ability to choose the point of cancellation may be especially important for future ODFD systems as currently the magic cancellation point for vibration does not overlap with the cancellation point for temperature fluctuations in our fiber SBS resonator. However, should it be possible to overlap both cancellation points, a vibration and temperature insensitive microwave synthesizer may be achievable.”

- 3. In the implementation of SBS lasers, fiber cavities of 10 meters in length are used, resulting in a mode spacing of 20 MHz, while the typical SBS gain bandwidth is 50 MHz. I am wondering whether such a long fiber might induce mode hopping in the SBS lasers under vibration conditions. If this occurs, it could alter the two beat notes produced by the beating of the SBS lasers and their nearest comb lines, which might lead to the failure of the phase locking of DRO.*

That is a very perceptive question raised by the reviewer. Indeed, the 10-m length used here is near the maximum of what can be employed without risking significant multimode effects in the SBS resonator. At lengths of 10 meters, we haven't encountered a second mode in the vicinity of our operation point yet, and in general we haven't experienced instability in the phase locking operation. This may be because, even though the mode spacing of 20 MHz is within the ~50 MHz SBS gain bandwidth, the gain bandwidth sets only the 3 dB from peak point in the gain profile. Modes that are 20 MHz apart will still experience lower gain, which appears to prevent them from lasing in our case. We would caution against going to longer lengths than 10 meters though, as we expect those configurations may have only specialized operation windows where the apparatus is stable. We have updated Methods C with the following:

“The mode spacing is 20 MHz, which is smaller than the SBS gain bandwidth of ~50 MHz. This gain bandwidth only specifies the point where the gain is attenuated by 3 dB, and we have verified that our current 20 MHz mode spacing yields a sufficient gain differential to enable stable single-mode operation of the SBS laser.”

- 4. From Fig. 4b, the OFD system exhibits relatively high phase noise at offset frequencies below 10 Hz, which I believe is primarily dominated by the SBS fiber lasers. Could the authors provide further details on the noise sources of the SBS lasers in this frequency range and discuss potential methods for noise reduction?*

We thank the reviewer for this suggestion. At low frequencies, the slope of the phase noise curve is 30 dB/decade, which is above the ideal 20 dB/decade expected for white frequency noise. We believe this increase in slope is due to thermorefractive noise, which has a well known characteristic shape that is non-white in frequency noise and increasing at low offset frequencies. That our noise is limited by thermorefractive noise at lower offset frequencies makes sense upon comparing the delta SBS laser curve of Fig. 4b and the curve of a single SBS fiber laser. We find that the noise of the SBS laser difference greatly suppresses the noise at low frequencies, as expected for thermal noise common to both lasers derived from the same cavity. However, this noise suppression has a limit to its effectiveness, given that the condition for the magic cancellation point for vibration is currently different than that for temperature.

To improve on this noise, ideally one would engineer the cavity to have both the temperature and vibration cancellation points overlap. However, in the interim, the thermal noise may be improved on by increasing the baseline division factor or the cavity mode volume to average away thermal fluctuations. Typically, an increase in cavity mode volume is accomplished by increasing the cavity length, though at 10 meters, our cavity is near the maximum length possible for single mode operation. One could potentially employ much larger (~kilometer) fiber lengths and forgo the use of SBS as a means for noise reduction. This has the advantage of bypassing the intrinsic noise of the SBS gain process, which is expected to be low but may become dominant once all common-mode noise processes are cancelled out. We have added the following to the manuscript:

“At 1 Hz, 10 Hz, and 100 Hz offset frequencies, the 10 GHz phase noise reaches phase noise levels of -42 dBc/Hz, -72 dBc/Hz, and -102 dBc/Hz, respectively. For these low offset frequencies, the phase noise follows a slope of -30 dB/decade, which we believe to be still limited by thermorefractive noise. Though common-mode thermal noise is greatly attenuated in the SBS difference, the level of suppression is finite and presents a phase noise floor to the 10 GHz divided output. This low-frequency noise may be further reduced by using larger resonator mode volumes or greater baseline division factors.” ... “However, should it be possible to overlap both cancellation points, a vibration and temperature insensitive microwave synthesizer may be achievable. Such a scheme is also expected to improve the ODFD's current low-frequency thermorefractive noise limitations.”

5. *Given the application of the magic cancellation point, to what extent can enclosing the SBS resonator within a vibration-isolation housing further enhance phase noise cancellation under strong vibration conditions?*

We would like to thank the reviewer for the excellent question. As seen from Fig. 4a, the magic cancellation point operates by taking the base vibration sensitivity of the resonator and reducing that level via a process of common-mode noise cancellation. This means that if the vibration sensitivity of the resonator can be improved, the final vibration response after common-mode cancellation will also be improved by the same amount. This is true because while improving the vibration sensitivity of the resonator may attenuate the levels that which the modes fluctuate in response to a perturbation, the magic cancellation point acts

independently by taking the response – no matter how big or small – and cancelling the net effect on the difference frequency. We have modified a sentence in the manuscript to address this point directly:

“Furthermore, the acceleration suppression of the magic cancellation point uniformly reduces the acceleration response across the spectrum of the base laser with no sign of saturation, suggesting that further advancements are possible with improvements to the vibration resiliency of the base laser and/or to the degree of vibration suppression via the magic cancellation point.”

6. *What is the locking bandwidth of the PLL used to lock the DRO in the OFD system?*

The locking bandwidth of the DRO phase locked loop servo is ~500 kHz. We have added the following sentence to the manuscript:

“The servo bump results in a slight increase in noise of the 10 GHz divided output near the DRO phase locked loop bandwidth (~ 500 kHz).”

7. *From the measurement results in Fig. 3, why is the acceleration sensitivity highest in the Z-direction and lowest in the X-direction? What factors might contribute to this behavior?*

This is a good question, and we have some thoughts on the reasons for the differences between the various directions. The Z-direction (out of the plane) stands out from the other two directions as this is the direction most natural for vibrations to transmit into the resonator. The resonator, for example, sits on a vibration platform, which itself sits on an optical table. An acceleration applied in the Z direction, directly leads to a vertical displacement of the resonator. For the other two directions, the walls of the resonator enclosure are not in direct contact with the vibration table, and thus vibrations are only indirectly transmitted from the table to X and Y perturbations of the resonator. As for why there is an asymmetry between X and Y, we can only guess that it may be related to how the wiring and optical fiber in/out is set up in our system. When the box displaces in X or Y, there is a slight opposing force due to the way the fibers/wiring are anchored to the vibration platform. It is not clear to us currently if this helps or hinders the sensitivity to vibration, but we believe this does contribute to the asymmetry in the response between X and Y. We have added the following to the description of Fig. 3c and 3d:

“Across the spectrum, the acceleration sensitivity is consistently largest in the Z direction. We believe this to be due to the resonator sitting on the vibration platform along the Z axis. Table vibrations directly translate to perturbations of the resonator, whereas the other two directions do not have a direct point of contact to the table. Furthermore, the asymmetry between X and Y is likely a result of the positioning and anchoring of the optical fibers that exit the resonator enclosure.”

8. *Since the authors used a DRO to synthesize the 10 GHz signal, and Fig. 4c shows that the DRO exhibits relatively poor acceleration sensitivity, I wonder whether the DRO might lose phase locking under strong vibration conditions. If this is the case, an RF oscillator with reduced vibration sensitivity would also be necessary in the OFD system to enhance robustness under strong vibration conditions.*

That is a good point regarding the DRO's poor acceleration sensitivity. It is true that phase locked loops have strict requirements for noise performance in order to be able to successfully stay locked. Under cases of strong vibration, the DRO may exceed these limits. We have extended our Methods B written analysis with the following:

"In principle, every element in the OFD system is vibration sensitive. However, certain elements such as the pump lasers and the DRO are locked and greatly diminish the influence of vibration on their operation. This applies to the entire pathway of the pump light up to the point that the pump light converts to an SBS wave in the resonator. Note that for strong vibrations, the DRO may lose phase locking, and the DRO's acceleration sensitivity would then be an important consideration."

Reviewer 3:

1. *This work proposes a method for achieving maximum suppression of vibration noise in microwave signal synthesis using optical frequency division (OFD) and the principle of "magic cancellation point". The authors demonstrated a 22.6 dB vibration noise cancellation effect through theoretical derivation and systematic experimental verification, maintaining excellent phase noise performance at a carrier frequency of 10 GHz. Overall, this manuscript is innovative and persuasive in terms of experimental verification, and the theme has a potentially significant impact on microwave photonics, precision measurement, and communication systems. However, several significant issues related to the fundamental principle, experimental design, and conclusion need to be addressed prior to acceptance.*

We thank the reviewer for their generous assessment and careful review of our work.

2. *The manuscript emphasizes that only the SBS laser subsystem is placed on the vibration platform, while the entire OFD system is not tested under the same vibration environment. Therefore, the evaluation of the system's overall vibration resilience is limited. How would the full system behave under vibration? Would it be possible to achieve the same order of magnitude in vibration suppression for the complete setup? Is there a realistic potential for full-system vibration-resilient applications in the future?*

The reviewer brings up an excellent point regarding the limitations of our vibration testing. Although we only directly applied vibration to the SBS laser system, one may extract information on how the whole system would perform under vibration based on educated guesses and on knowledge of the OFD system and its stabilization loops. Methods Section B

provided an initial assessment for the reasons why the whole system would not perform very differently in response to vibration. This reasoning was based on a few observations: 1) The EOM comb system is exactly common to the comb line beat notes with both SBS lasers and exactly cancels when their difference frequency is taken. We have tested this by directly applying large frequency modulations to the central laser of the EOM comb and observed that the SBS difference remains constant. 2) The DRO that controls the RF modulation frequency and comb line spacing is vibration sensitive, but is ultimately locked to complete the frequency division process. For our single integrator servo with ~500 kHz bandwidth, the phase locked loop gain at frequencies where vibrations occur (<1 kHz) would be >54 dB, increasing at a rate of 20 dB/decade. This means that at 10 Hz offset, the total servo gain would be 94 dB, which can effectively combat large increases in the DRO phase noise that would otherwise occur in the presence of vibration. However, for large enough vibrations, the DRO may still lose phase lock, and a less vibration sensitive oscillator may be required. 3) All the rest of the oscillator frequencies used are low (<100 MHz), whether for PDH locking or for offset locking of the phase locked loop, and thus suffer orders of magnitude lower absolute frequency shifts in response to vibration. From this analysis, we can infer that the system would not behave much differently if the whole system could be tested for acceleration sensitivity.

We have expanded Methods Section B to more directly address the reviewer's questions. In the interest of space, we do not reproduce the modified Methods Section B here.

- 3. In the PDH locking process, the authors should clarify whether the phase modulation signals (LO1 and LO2) are derived from the same local oscillator source. Based on my knowledge, operating two PDH loops with independent LO sources may degrade the ultimate common-mode noise rejection and could introduce additional differential-mode noise.*

We thank the reviewer for pointing this out. We do actually use two separate function generators for the two PDH locks. The two function generators are kept at different frequencies so that the two PDH locks do not interfere with each other when detected by a common photodiode. Indeed, the reviewer is correct that if the two PDH locks were instead derived from the same function generator, the vibration-induced shifts of the PDH lock and thus of the locked laser frequencies would cancel out when the difference frequency is taken. In our case, we find that because the function generator frequency is much lower than that of the optical frequencies to be divided down, the effects of vibration are greatly suppressed. Under cases of very strong vibration, the two function generators can be synchronized together from their RF in/out ports to make their responses to vibration common.

We have added these details into Methods Section B:

“Other elements, such as function generators, operate at low frequencies relative to the ~200 THz optical waves, and are far less susceptible to vibration. Note that all the function generators used here were operated independently from one another. However, these function generators may also be synchronized should they prove to be a limitation to the common-mode rejection.”

4. *In the EOM subsystem, the DRO output signal is distributed to multiple phase modulators via several phase shifters. The authors should provide detailed descriptions of the phase shifts applied to each branch because the proper phase alignment is critical for ensuring the uniformity of the comb teeth generated by the EOM system.*

The reviewer is indeed correct that the phase alignment of each of the EOM combs is critical to the shape and extent of the comb spectrum. This information however, isn't the easiest for us to obtain due to the fact that the required phase is dependent on the length of the RF cabling to each of the EOMs, any RF components along the way adding additional phase delay, and the optical fiber path length between EOMs. That is, the required RF phase for subsequent EOMs has to serve the role of aligning with the previously applied RF modulation, and so the EOM phase varies with the signal delay on both optical and RF paths. In practice, we simply have an analog knob on each of the phase shifters for the EOMs that is turned until the comb spectrum bandwidth is maximized. There isn't a readout for the applied phase that we are aware of.

We have added the following explanations to Methods Section C: "The EOM comb is intentionally operated with the seed laser frequency at the center between the two SBS laser outputs. To achieve the maximum comb span, the phases of the RF drive to each of the modulators need to be aligned for constructive interference. The exact phase required is dependent on the phase delay along both the RF and optical paths, and will be different system to system. As a consequence, these phases are manually tuned via analog control knobs while monitoring the comb extent on an optical spectrum analyzer."

5. *The identification of the "magic cancellation point" in the manuscript is primarily based on experimental observation. However, the theoretical analysis related to common-mode noise cancellation is overly simplified. The manuscript lacks supporting theoretical modeling and simulation to validate the underlying mechanism.*

We strongly agree with the reviewer that the theoretical analysis and modeling of the magic cancellation point is lacking in the manuscript. Part of the challenge with simulating the system is that the many degrees of freedom that govern the ways in which the fiber resonator can move results in a complex simulation environment that is highly dependent on how the fiber is wound. There are for example winding techniques of optical fiber that greatly suppress and cancel sensitivity to acceleration. Currently, our fiber resonators are wound on a cylindrical mandrel, and the winding pattern is left to relax to an unknown arbitrary state.

On the other hand, we feel that the theoretical analysis does have possibility to be greatly improved by continuing the derivation to relate the magic cancellation point to system parameters. We have added these details to Methods Section A. Equations (14)-(18) show how the condition for magic cancellation directly relates to the refractive indices and derivatives of refractive indices with respect to applied acceleration for the two resonator modes used as the baseline in optical frequency division. Due to the asymmetry between modes of different mode families, there will always be a magic cancellation point, but the baseline extent will depend

on the degree of mode asymmetry. In addition to the added derivations, we have also expanded the analysis of the magic cancellation point within Methods Section A. For space saving reasons, we do not duplicate the added derivations and analysis here.

6. *From the experimental results shown in Figure 3, it is evident that the vibration cancellation along the Z-axis is the least effective. The authors should explain why the Z-axis exhibits the worst cancellation performance compared to the X- and Y-axis.*

The reviewer brings up an important question regarding the Z-axis acceleration response being larger than that of the other two axes. We have many thoughts as to why this might be the case for our system. The Z-direction (out of the plane) stands out from the other two directions as this is the direction most natural for vibrations to transmit into the resonator. The resonator, for example, sits on a vibration platform, which itself sits on an optical table. An acceleration applied in the Z direction, directly leads to a vertical displacement of the resonator. For the other two directions, the walls of the resonator enclosure are not in direct contact with the vibration table, and thus vibrations are only indirectly transmitted from the table to X and Y perturbations of the resonator. As for why there is an asymmetry between X and Y, we can only guess that it may be related to how the wiring and optical fiber in/out is set up in our system. When the box displaces in X or Y, there is a slight opposing force due to the way the fibers/wiring are anchored to the vibration platform. It is not clear to us currently if this helps or hinders the sensitivity to vibration, but we believe this does contribute to the asymmetry in the response between X and Y. We have added the following to the description of Fig. 3c and 3d:

“Across the spectrum, the acceleration sensitivity is consistently largest in the Z direction. We believe this to be due to the resonator sitting on the vibration platform along the Z axis. Table vibrations directly translate to perturbations of the resonator, whereas the other two directions do not have a direct point of contact to the table. Furthermore, the asymmetry between X and Y is likely a result of the positioning and anchoring of the optical fibers that exit the resonator enclosure.”

7. *In Figure 4, the phase noise data is presented only under static conditions without applied vibration. The manuscript lacks a comparative measurement of phase noise under vibration, both with and without magic cancellation applied. Such a comparison is necessary to demonstrate the effectiveness of the magic cancellation mechanism in improving the final signal quality under realistic vibration conditions.*

We thank the reviewer for providing this suggestion. The applied acceleration is performed by applying a known sinusoidal modulation to the vibration platform, and measuring the phase noise response of the ODFD system to the sinusoidal modulation across a range of frequencies. Due to the relatively low transduction of the applied voltage to acceleration of the table, we needed to not only max out the voltage of our applied signal but also to concentrate all of the vibration energy into one peak (hence the sinusoidal modulation). We have added the phase noise spectra of the ODFD system with and without operation at the magic cancellation point

to the Supplementary Information. The magic cancellation point is turned off by switching the polarization of the second laser to match that of the first laser. Furthermore, in anticipation of the reviewer's point 9, we have also added the concurrently measured acceleration response to the set of figures. These figures can be found in the Supplementary Information Figure 2.

8. *According to Figure 2, two beat notes from a highspeed photodetector are split equally into two ways and then mixed together. The structure is confusing because the two signals injected into the RF mixer are identical and the sum or difference frequency cannot be obtained at output of the mixer. Clarifications or corrections should be made to this part.*

We greatly appreciate the reviewer's attentiveness to detail and for their careful checking of our work. This configuration is indeed very strange as after the photodetector, the signal path appears to split in two and then are mixed together. This would seemingly yield either zero or twice the signal's RF signal frequency at the output of the mixer. The key to understanding this configuration is to approach the system from the standpoint that the photodetector output consists of two signals, which are the two beat notes of the SBS laser with their adjacent comb lines. Both beat notes are present on each of the signal paths when the photodetected output is split in two. When the two paths are mixed, six signals are generated: 1) $RF_1 - RF_1$, 2) $RF_1 + RF_1$, 3) $RF_2 - RF_2$, 4) $RF_2 + RF_2$, 5) $RF_1 - RF_2$, 6) $RF_1 + RF_2$. The first four of these mixing products are either zero frequency or twice the RF frequency for one of the beat notes, and thus are irrelevant to the ODFD scheme. The last two mixing products provide the desired sum or difference signals that can be used for phase-locking the DRO. Note that the system is set up this way in order to maximize the amount of common pathway shared by the two SBS laser signals, all the way up to the photodetector and RF splitter.

We have added the following to the discussion of Figure 2:

“After photodetection on a high speed photodetector, the two beat notes (BN 1 and BN 2) are split equally two ways and then mixed together. The mixing products consist of the following: $BN_1 - BN_1$, $BN_1 + BN_1$, $BN_2 - BN_2$, $BN_2 + BN_2$, $BN_1 - BN_2$, and $BN_1 + BN_2$. Only $BN_1 - BN_2$ and $BN_1 + BN_2$ are relevant for use in ODFD phase locking. Depending on the polarity of the beat note between the nearest comb line and the SBS lasers, either the sum or difference frequency should be selected after mixing”

9. *The author should add key vibration parameters in the acceleration sensitivity measurement, such as the vibration power spectral density or the acceleration value.*

We thank the reviewer for this excellent suggestion. We have addressed this part in our response to the reviewer's point 7, as the two points are related.

10. *In this paper, the proposed system is phase locked to an RF reference using the ODFD method. The comparison of phase noise and acceleration sensitivity in Fig. 4(c) is not appropriate, as the oscillators provided in literature [28] operate in a free-running state without any phase-*

locking loop. This discrepancy could result in incorrect conclusions, and the comparison should be carefully revised.

We thank the reviewer for pointing this out and agree with the reviewer that the comparison in Fig. 4c isn't completely fair. We think that the main contention point is in the comparison of phase noise across the various oscillators, as the phase locked loop servo does not alter the acceleration sensitivity outside of enabling ODFD at the magic cancellation point. The equivalent of PDH locking does exist for microwave oscillators, but appears to be rarely employed. The main area where resonator Pound locking is employed appears to be sapphire loaded cavity oscillators. From the reported results of L. Kensbock et al., "A 20 GHz Sapphire-Loaded Cavity Oscillator with Pound Locking and Highest Phase Stability at Room Temperature," IEEE 2024 Asia-Pacific Conference on Microwave – Pound locking does improve the oscillator's phase noise, but the improvement does not appear to be large. On the other hand, phase locked loops are commonly employed, but these servos simply stabilize one oscillator to another without any further improvements to phase noise.

We find it challenging to strike a balance between providing some comparison across microwave oscillators to the readers and making certain that the comparison is completely valid. To remedy this, we have opted to provide explanations in the main text and also in the figure captions to make it clear to the reader that the ODFD system comprises multiple servos. Thus, the performance of the ODFD system cannot be strictly compared to free-running microwave oscillators. We have removed the sentence "Our ODFD combines both excellent phase noise and acceleration performance" to get rid of references of a direct comparison between the ODFD and the other free-running oscillators. This will hopefully prevent a situation where the reader adopts an incorrect conclusion from the manuscript.

The figure caption now reads "Comparison of phase noise and acceleration sensitivity across several free-running microwave oscillator technologies at 10 GHz. STW, surface transverse wave; OEO, optoelectronic oscillator; ACCRO, air-dielectric ceramic-cavity resonator oscillator; OCXO, oven controlled crystal oscillator; SLCO, sapphire loaded cavity oscillator. Our phase-locked ODFD performance is also plotted."

The main text now reads: "Figure 4c showcases the phase noise and acceleration sensitivity of the ODFD system and of a variety of microwave oscillators in existence today. Note that the microwave oscillators all operate free-running, which cannot be strictly compared against the phase-locked operation of the ODFD. Nevertheless, with these differences in mind, it is still useful to put the ODFD system performance in the context of commonly used microwave oscillators. For the purposes of uniformity, phase noise is compared scaled to a center frequency of 10 GHz and at an offset frequency of 10 Hz, while the acceleration sensitivity is compared across oscillators using the axis most sensitive to acceleration."